# Pyrroloquinoline Quinone Alleviates Intestinal Inflammation and Cell Apoptosis via the MKK3/6-P38 Pathway in a Piglet Model

**DOI:** 10.3390/ijms25179723

**Published:** 2024-09-08

**Authors:** Caiyun Huang, Xuanci Yu, Ziyuan Du, Zhihao Zhu, Chenyu Shi, Ang Li, Fenglai Wang

**Affiliations:** 1College of Animal Science, Fujian Agriculture and Forestry University, Fuzhou 350002, China; 15611021795@163.com (X.Y.); 13266460005@163.com (Z.D.); moonhalo_703@163.com (Z.Z.); poul@fafu.edu.cn (A.L.); 2State Key Laboratory of Animal Nutrition, College of Animal Science & Technology, China Agricultural University, Beijing 100193, China; scyshichenyu@163.com (C.S.); wangfl@cau.edu.cn (F.W.)

**Keywords:** pyrroloquinoline quinone, intestine, inflammation, apoptosis, weaned piglets

## Abstract

This study investigates the underlying mechanism through which dietary supplementation of pyrroloquinoline quinone disodium (PQQ) alleviates intestinal inflammation and cell apoptosis in piglets challenged with lipopolysaccharide (LPS). Seventy-two barrows were divided into three groups: control (CTRL), LPS challenged (LPS), and LPS challenged with PQQ supplementation (PQQ + LPS). On d 7, 11, and 14, piglets received intraperitoneal injections of LPS or 0.9% of NaCl (80 μg/kg). After a 4 h interval following the final LPS injection on d 14, blood samples were obtained, and all piglets were euthanized for harvesting jejunal samples. The results showed that dietary supplementation of PQQ improved the damage of intestinal morphology, increased the down-regulated tight junction proteins, and reduced the increase of serum diamine oxidase activity, the intestinal fatty acid binding protein, and TNF-α levels in piglets challenged with LPS (*p* < 0.05). The proteomics analysis revealed a total of 141 differentially expressed proteins (DEPs), consisting of 64 up-regulated DEPs and 77 down-regulated DEPs in the PQQ + LPS group compared to the LPS group. The KEGG pathway analysis indicated enrichment of the tight junction pathway and the apoptosis pathway (*p* < 0.05). Compared to the LPS group, the piglets in the PQQ + LPS group had increased levels of Bcl-2 protein, reduced positive apoptosis signals, and a decrease in the abundance of MKK 3/6 and p-p38 proteins (*p* < 0.05). In conclusion, dietary supplementation of PQQ could alleviate jejunal inflammatory damage and cell apoptosis in piglets challenged with LPS through the MKK3/6-p38 signaling pathway.

## 1. Introduction

The small intestine plays a crucial role in nutrient digestion and absorption, while also serving as the primary defense barrier against pathogens and endotoxins [1]. However, inflammation triggered by external harmful agents can lead to intestinal damage, diarrhea, and growth retardation in young animals [2]. Persistent inflammation can stimulate intestinal epithelial cells to produce an abundance of proinflammatory cytokines [3]. These cytokines have the ability to activate MAP Kinase Kinase (MKK) 3/6, resulting in the phosphorylation of p38. Subsequently, this phosphorylation event promotes cytochrome c synthesis, increases caspase-3 expression, and ultimately induces apoptosis in intestinal epithelial cells, leading to damage to the small intestinal mucosal barrier function [4]. Therefore, by modulating the production of inflammatory cytokines and intervening in cell apoptosis, positive effects can be achieved in maintaining the integrity of the intestinal barrier function.

Pyrroloquinoline quinone (PQQ) has been recognized as a novel cofactor of microbial dehydrogenases [5]. It is an orthoquinone compound known for its high redox potential, enabling it to undergo reduction by two electrons [6]. Under neutral to mildly acidic conditions, PQQ primarily exists as an anion due to its acidic functional groups. This anionic form facilitates the formation of reversible covalent adducts with various compounds such as water, cyanide, ammonia, hydrazine, ethanol, aldehydes, and ketones [7]. PQQ has been proven to promote growth performance in broilers [8], weaned rats [9], and piglets [10]. Pretreatment with PQQ has been shown to alleviate inflammation and cell apoptosis in Kupffer cells [11] and in astrocytes [12] when exposed to LPS. Besides, PQQ supplementation can regulate jejunal inflammation and improve intestinal epithelial integrity in piglets [13] and in broiler chickens [14]. Previously, we have shown that dietary supplementation of 3.0 mg/kg PQQ can decrease feed to gain ratio [15] and alleviate jejunal mucosal inflammation induced by *Escherichia coli* in piglets [16]. What is more, porcine intestinal epithelial cells were incubated with 10 nmol/L PQQ that can inhibit hydrogen peroxide-induced cell apoptosis [17]. However, there is limited research available on the mechanisms associated with the regulation of intestinal inflammation and cell apoptosis through PQQ supplementation in piglets.

Therefore, our hypothesis is that a diet supplemented with PQQ could regulate intestinal inflammation and cell apoptosis through the MKK3/6-p38 pathway. In this study, we used a piglet model challenged with Escherichia coli LPS to assess whether dietary supplementation of PQQ can attenuate LPS-induced intestinal inflammation and cell apoptosis by modulating the MKK3/6-p38 signaling pathways.

## 2. Results

### 2.1. The Growth Performance

Throughout the 14-day feeding trial, the final body weight (BW) and ADG were decreased, and the diarrhea index was increased in the LPS group compared with the CTRL group (*p* < 0.05). No differences were observed in Initial BW, Final BW, ADG, ADFI, FCR, and the Diarrhea index between the LPS group and the PQQ + LPS group (Table 1).

### 2.2. The Serum and Jejunal Mucosal Factors Contents

Compared with the CTRL group, the serum DAO and I-FABP activities, mucosal D-LA, TNF-α, and the IFN-γ concentration were increased, and the the level of serum ALP was decreased in the LPS group (*p* < 0.05). In comparison to the LPS group, the serum DAO and I-FABP and mucosal TNF-α were reduced in the PQQ + LPS group (*p* < 0.05). However, there were no differences observed in mucosal I-FABP, TGF-β1, and IL-1β among the three groups (Table 2).

### 2.3. The Morphology and Tight Junction Proteins Expression in Jejunum

The CTRL group displayed an intact structure of jejunal villi, with a clear border of the central lacteal. The epithelial structure of the mucosal layer in the villi was complete and normal in appearance, with no detachment observed. The crypts in the mucosal layer were neatly and tightly arranged without expansion, the submucosa showed no signs of edema, and there was no significant infiltration of inflammatory cells within the tissue (Figure 1(A1–A3)). In the LPS group, the morphology of the jejunal villi was irregular, with noticeable epithelial detachment in the mucosal layer, exposing the lamina propria. The crypts exhibited characteristics of edema, along with significant infiltration of inflammatory cells (Figure 1(B1–B3)). In contrast, the PQQ + LPS group exhibited distinct characteristics in the structure of villi and central lacteal, although some inflammatory cell infiltration was still observed (Figure 1(C1–C3)). In addition, compared to the control group, the villus height and crypt depth ratio were significantly reduced in the LPS group. PQQ supplementation mitigated these reductions, increasing the villus height and crypt depth ratio that had been decreased by the LPS treatment. There were no significant differences in villus height and crypt depth among the three groups (Figure 1D).

Compared with the CTRL group, the protein abundance of ZO-1 and Occludin in jejunum were decreased in the LPS group (*p* < 0.05). The protein abundance of ZO-1 and Occludin in the jejunal mucosa of the LPS + PQQ group was significantly higher compared to the LPS group (*p* < 0.05). However, no significant differences were observed in ZO-3 protein expression among the three groups (*p* < 0.05, Figure 2).

### 2.4. Analysis of Proteomic Profile in the Jejunal Mucosa

A total of 110,724 spectra were identified, resulting in the characterization of 35,696 peptides. These peptides corresponded to the identification of 5321 quantified proteins. Among these proteins, 141 differentially expressed proteins (DEPs) were obtained (Appendix A). Specifically, 64 DEPs were up-regulated and 77 DEPs were down-regulated in the PQQ + LPS group compared to the LPS group (Figure 3A,B). To validate the accuracy of the proteomic analysis results, four differentially expressed proteins (DEPs) were randomly selected for validation using the Western blot and ELISA techniques. These DEPs include MAP Kinase Kinase 3/6 (MKK3/6, UniProtKB accession no. F1RV28), protein kinase R-like ER kinase (PERK, UniProtKB accession no. K7GP80), Immunoglobulin G (IgG, UniProtKB accession no. L8B139), and amine oxidase (UniProtKB accession no. F1S1G8). The findings from these validation experiments confirmed the high reliability of the proteomic analysis results (Figure 3C–E).

In GO function annotation, 39 significantly clustered biological processes were selected, of which 24 terms were for biological processes (BP), 6 terms for cellular component (CC), and 9 terms for molecular function (MF) (Figure 4). These terms had at least one DEPs and were involved in 5 binding terms, 11 immunologic process terms, 4 kinase activity terms, 13 metabolic process terms, and 6 mitochondrial structure and function terms.

In the KEGG pathway analysis (Table 3), the antigen processing and presentation pathway showed enrichment of three DEPs: MHC I6, MHC I2, and Cathepsin L1. Additionally, two DEPs were enriched in both the apoptosis pathway and the intestinal immune network for the IgA production pathway, namely IgG heavy chain and Poly-Ig receptor. The Fc epsilon RI signaling pathway exhibited enrichment of two DEPs: IgG heavy chain and MKK3/6. Three DEPs were enriched in the Inflammatory mediator regulation of TRP channels pathway, including MKK3/6, Cytochrome P450 2C42, and Calcium/calmodulin-dependent protein kinase. The Tight junction pathway showed enrichment of MAGUK p55 subfamily member 5, Claudin, and PATJ crumbs cell polarity complex component.

### 2.5. MKK3/6-p38 Signaling Pathway and Apoptosis

Compared to the CTRL group, the levels of p-p38 and Caspsase-3 protein increased, and the expression of Bcl-2 protein decreased in the LPS group (*p* < 0.05). The increased p-p38 and Caspsase-3 protein and the decreased Bcl-2 protein were attenuated in the PQQ + LPS group compared with those in the LPS group (*p* < 0.05, Figure 5). No significant difference was found for p38 and Bax protein in the jejunal mucosa among the three groups (*p* > 0.05).

Further analysis of jejunal apoptosis levels using TUNEL staining revealed that the fluorescence intensity of apoptosis-positive cells in the jejunum was higher in piglets challenged with LPS compared to the CTRL group. However, in the PQQ + LPS group, the fluorescence intensity was weak compared to the LPS group (Figure 6).

## 3. Discussion

Bacterial endotoxin attacks often lead to intestinal barrier dysfunction, diarrhea, and growth retardation in young animals. LPS, a potent endotoxin, was used to establish a model of intestinal inflammatory injury [18,19]. PQQ, as a vitamin-like substance, possesses multiple beneficial characteristics, including growth promotion [20], antioxidant [21], anti-inflammatory [22] and anti-apoptosis functions [23]. In our previous study, we have shown that PQQ supplementation regulates jejunal mucosal inflammatory damage in piglets challenged with Enterotoxigenic *E. coli* [16] and decreases porcine intestinal epithelial cell apoptosis induced by hydrogen peroxide [17]. However, the molecular mechanisms underlying how PQQ regulates intestinal cell apoptosis induced by inflammatory damage in piglets remain unclear. Additional investigation is required to clarify these mechanisms and gain a comprehensive understanding of the distinct pathways and signaling molecules implicated in the protective properties of PQQ on the small intestine.

In our previous reports, diet supplementation with 3.0 mg/kg PQQ improved ADG and FCR and reduced the diarrhea index in piglets [16,24]. Nevertheless, in the present study, we did not observe improvements in growth performance or diarrhea index in piglets challenged with LPS when supplemented with PQQ. The limited sample size of six replicates in this study may have contributed to this difference in results. Further research with a larger population of piglets is warranted to explore the potential effects of PQQ supplementation on growth performance under LPS challenges.

Maintaining the integrity of the intestinal barrier is crucial to prevent the penetration of luminal bacteria and endotoxins into the mucosa [25]. Intestinal health, particularly intestinal integrity, can be assessed by examining intestinal morphology and the intestinal barrier function. Intestinal morphology, which includes villous height, crypt depth, and VCR, directly indicates the state of the intestines [26,27,28]. The intestinal barrier function can be evaluated through various serum parameters, such as the activities of DAO and I-FABP. When the integrity of the intestinal mucosal barrier is compromised, intestinal DAO and I-FABP are released into the bloodstream, leading to increased plasma DAO [29] and I-FABP activity [30]. In our study, we observed elevated serum DAO and I-FABP activity in piglets challenged with LPS, indicating a compromised intestinal barrier function. Previous studies have shown that the LPS challenge induces intestinal inflammation in piglets, as evidenced by increased levels of inflammatory factors such as IFN-γ, TNF-α, and IL-1 in the mucosa [31]. Intestinal inflammation further contributes to the impairment of the intestinal barrier function, as observed through incomplete intestinal morphology and reduced expression of tight junction proteins [32]. Two crucial members of intestinal tight junction proteins, ZO-1 and Occludin, play a vital role in regulating the structural composition and permeability of the intestinal epithelium [33]. In our research, we observed changes in jejunal morphology, increased mucosal contents of IFN-γ, and TNF-α, as well as decreased levels of ZO-1 and Occludin proteins. These findings suggest that LPS-induced intestinal barrier function inflammatory injury occurs in weaned piglets. Therefore, we effectively established an intestinal inflammatory injury model in piglets challenged with LPS.

We showed that dietary supplementation of PQQ to LPS challenged piglets effectively restored the change of intestinal morphology and decreased the serum DAO and jejunal inflammatory factors. These results are also consistent with previous findings, in which PQQ can alleviate the intestinal inflammatory injury in piglets [13]. Inflammatory response in the intestine can readily destroy epithelial permeability and integrity and induce cells apoptosis [34], as evidenced by a decrease in the expression of tight junction proteins and Bcl-2/Bax ratio, and the translocation of activated caspase-3 from mitochondrial Cyt-c to the cytoplasm. PQQ could regulate the expression of Bcl-2, Bax and caspase-3 to alleviate the cell apoptosis in rat Kupffer cells on LPS-induced inflammatory damage [11]. A diet supplemented with PQQ can enhance anti-apoptotic effects by in chick fatty liver disease [23]. In our previous research, pretreatment with PQQ decreased apoptosis-related protein and increased the tight junction proteins in porcine small intestinal epithelial cells challenged with hydrogen peroxide [17]. In the present study, the proteomic analysis results revealed that the tight junction pathway and apoptosis pathway were enriched by DEGs to the GO annotation and the KEGG database. Dietary supplementation of PQQ modulates the apoptotic-positive signal, the proteins expression of Bcl-2, caspase-3, ZO-1 and Occludin under LPS stimulation in jejunum tissues. These results suggest that PQQ has the potential to protect the integrity of the jejunal mucosal barrier and prevent apoptosis of intestinal epithelial cells induced by LPS.

Notably, among the DEPs, the MKK3/6 protein is a bispecific kinase belonging to the MKK kinase family. This protein can be activated by various stimuli, including endotoxin and inflammatory factors and subsequently transmits a signal to phosphorylate p38 [4]. This signaling cascade ultimately triggers cell apoptosis [35]. The activation of the MKK3/6-p38 pathway has been reported to promote apoptosis in intestinal epithelial cell induced by LPS [36] and neuroblastoma cells challenged with Aeromonas hydrophila cytoenterotoxins [37]. In the present study, a diet supplemented with PQQ decreased the evaluation of MKK3/6 and the p-p38 protein. Therefore, we reasonably deduced that PQQ inhibited the progress of cell apoptosis in weaned piglets challenged with LPS, which is associated with the inhibition of the MKK3/6-p38 pathway.

## 4. Materials and Methods

### 4.1. Ethics Approval

All experimental protocols used in the present study were approved by the Institutional Animal Care and Use Committee of Fujian Agriculture and Forestry University (ID: PZCASFAFU24002).

### 4.2. Animals and Experimental Design

Seventy-two crossbred barrows (Duroc × Landrace × Yorkshine), with an initial body weight of 7.16 ± 0.85 kg and at the weaned age of 27 ± 2 d, were randomly divided into three groups. Each group consisted of six replicates, with four piglets per replicate (pen). Piglets were housed in a 1.80 × 1.10 m pen in a well-ventilated, environment-controlled facility maintained at 29 ± 2 °C. All piglets had ad libitum access to feed and water. The study consisted of two dietary treatments: a basal diet without antibiotic additives and a basal diet supplemented with 3 mg/kg pyrroloquinoline quinone disodium. The PQQ·Na₂ (purity ≥ 98%) was synthesized, its purity verified, and generously donated by the Changmao Biochemical Engineering Company (Changzhou, China). Before incorporation into the diet, it was diluted with corn starch to achieve a concentration of 1 g/kg mixture. The nutritional levels of all experimental diets complied with the requirements outlined in the National Research Council (NRC) 2012 guidelines [24] (Appendix A). The dosage of PQQ used in this study was described in our previous study [16].

The methodology for establishing the model of piglets challenged with LPS has been described in our previous research [26]. Briefly, on day 7, 11, and 14 of the experiment, piglets from two groups were offered intraperitoneal injection of LPS (*E. coli* serotype 055:B5; Sigma, St. Louis, MO, USA), 80 µg/kg in body weight, and piglets from the control group received an equivalent amount of 0.9% of NaCl. Therefore, three treatments were included in this study: (1) piglets that received a basal diet and were injected with 0.9% of NaCl (CTRL group); (2) piglets that received a basal diet and were challenged with LPS (LPS group); (3) piglets that received a 3.0 mg/kg PQQ diet and were challenged with LPS (LPS + PQQ group).

### 4.3. Growth Performance and Sample Collection

In this experiment, the weights of piglets were measured on day 14 to calculate the average daily gain (ADG), average daily feed intake (ADFI), and feed conversion ratio (FCR). Diarrhea incidence was recorded by a single observer throughout the experiment and was assessed using the following scale: well-formed feces = 1, unformed feces = 2, and watery feces = 3. The diarrhea incidence for piglets in each pen was calculated as follows:[(number of piglets with watery feces × number of days of diarrhea)/(total number of piglets × number of days in the experiment)] × 100%.

Four hours after the last LPS injection on day 14, blood samples were collected from all piglets to obtain the serum. Subsequently, the piglets were euthanized, and a 2 cm segment of the intestinal tissue was fixed in 4% neutral-buffered paraformaldehyde for morphological examination. Additionally, another 2 cm segment from the mid-jejunum was rapidly frozen in liquid nitrogen and stored at −80 °C for subsequent analysis.

### 4.4. Intestinal Morphology Analysis and TUNEL Staining

To analyze the morphology of the jejunal tissue, hematoxylin and eosin (HE) staining was performed following the protocol described in our previous research [38]. Briefly, after 24 h of fixation in a 4% paraformaldehyde solution, the intestinal segments underwent several steps: dehydration, embedding in paraffin, slicing into approximately 5 μm thick sections, and staining with HE. In each treatment group, one slide was prepared for each pig, resulting in six slides with two separate sections each. At least 10 well-oriented villi and crypts were measured on each slide, totaling a minimum of 60 replicate locations per group. Villus height was measured from the tip of the villus to the villus-crypt junction, while crypt depth was defined as the depth of the invagination between adjacent villi. The villous height, crypt depth, and the villous height to crypt depth ratio were calculated following procedures outlined in our previous study [39].

For the analysis of DNA fragmentation in apoptosis, the TUNEL assay was employed based on a previous study [27]. In short, the deparaffinized sections were fixed and permeabilized. Subsequently, the sections were incubated with the TdT Reaction mixture at 37 °C for 1 h. Following this, the slides were incubated with the Click-iT Plus TUNEL reaction cocktail at 37 °C for 30 min and then stained with DAPI.

The histological pathological features and the fluorescence intensity of apoptosis-positive cells in the jejunum were assessed using a light microscope (Olympus BX-51, Olympus Corporation, Tokyo, Japan).

### 4.5. Serum and Mucosal Cytokines

The levels of D-Lactic acid (D-LA) and intestinal fatty acid binding protein (I-FABP) in both serum and mucosal samples, as well as the serum level of alkaline phosphatase (ALP), and the mucosal in IL-1β, TNF-α, the transforming growth factor (TGF)-β1 and interferon gamma (IFN-γ) (eBioscience, San Diegom, CA, USA) and diamine oxidase (DAO) activity in the serum (Nanjing Jiancheng Bioengineering Institute, Nanjing, China) were detected by the ELISA method using commercial swine kits on the basis of product manuals. The absorbance of DAO was detected at 460 nm, and the others were measured at 450 nm. The minimal detection limit was 1.28 ng/mL for DAO, 0.04 mmol/L for D-LA, 40 U/L for ALP, 2.21 pg/mL for I-FABP, 0.3 pg/mg for IL-1β, 5 pg/mg for TNF-α, and 8.23 pg/mg for TGF-β1, respectively.

### 4.6. Western Blot Analysis

Protein expression levels in jejunal mucosa were measured following the protocol described in our previous research [17]. GAPDH or β-actin was used as a reference gene for normalization and as a control. In brief, 100 mg of frozen jejunal mucosa was powdered and homogenized in RIPA buffer. The homogenate was centrifuged to collect the supernatant which was used to measure protein concentration using the BCA method [40]. A total of 80 µg of proteins from each sample were separated on 10% SDS polyacrylamide gels and transferred onto PVDF membranes. The membranes were then blocked at room temperature and incubated with the primary antibodies overnight at 4 °C. Details of the specific primary antibodies are listed in Appendix A. Next, membranes were incubated by a DyLight 800-conjugated secondary antibody for 1 h. The band densities were read by the Odyssey Clx system (LI-COR Biotechnology, Lincoln, NE, USA) and quantified by Image J software (version 2).

### 4.7. LC-MS/MS Experiments

Protein from jejunal mucosa was digested by the FASP method [41]. Briefly, the 100 µg protein extracted from jejunal mucosa underwent the following steps: the protein was mixed with a UA Buffer and incubated in the dark for 40 min with 55 mM IAA. Subsequently, the sample was digested and vacuum-dried. For the subsequent data-dependent acquisition (DDA) analysis, a total of 11 µg of peptides per sample were combined and fractionated offline using a High pH RP column (Thermo Scientific, Pierce^TM^ High pH Reversed-Phase Peptide Fractionation Kit, Waltham, MA, USA). The raw data for both DDA and data-independent acquisition (DIA) were obtained using the Q Exactive HF instrument in combination with the Easy nanoLC 1000 system (Thermo Scientific). The data quality analysis of DIA was conducted, and the findings unequivocally showcased the precision, dependability, and reproducibility of the data quality (Appendix A).

### 4.8. DDA Data Search and DIA Data Analysis

All the DDA raw data were uploaded and searched in the Proteome Discoverer (PD) software (Version 2.1, Thermo Fisher Scientific) and the Sus Scrofa fasta database (49,446 entries, downloaded on 27 May 2019), respectively. The parameters were set as follows: The miss cleavage site = 2; False discovery rate (FDR) = 1%. The library generated after the search result of DDA was imported to Skyline (4.2.0.18305). Next, the DIA raw data from six samples were quantified. The mProphet peak scoring model was used to all detected peaks reintegration, per dataset, and iteration was analyzed independently. Each peak was matched for a q-value. Then, for further bioinformatics analysis, results from the protein quantitation were exported by MSstats 23. The mass spectrometry proteomics data were deposited to the ProteomeXchange Consortium (https://proteomecentral.proteomexchange.org, accessed on 3 April 2024) via the iProX partner repository with the dataset identifier PXD051166.

### 4.9. Statistical Analysis

Statistical analysis of data obtained from the diarrhea rate was analyzed by the Chi-square test for the value of growth performance. Serum and intestinal mucosal factors expressions were evaluated through one-way ANOVA. Specifically, statistical differences among mean values were assessed using Tukey’s multiple range test. The validation of proteomics data was analyzed by a *t*-test (SAS, version 9.2).

The protein abundance data obtained from DIA proteomics was subjected to log_2_ transformation and evaluated using the mapDIA software (https://github.com/mapDIA/mapDIA, accessed on 5 November 2022). Differentially expressed proteins (DEPs) were identified based on the criteria of *p* < 0.05 and |log_2_FC| > 1. Goatools software (https://github.com/tanghaibao/goatools, accessed on 11 November 2022) and KOBAS (2.0) analyses were utilized to identify significantly enriched Gene Ontology (GO) terms and Kyoto Encyclopedia of Genes and Genomes (KEGG) pathways. The statistical significance of the enrichment was determined using Fisher’s exact test. Differences were considered significant when *p* < 0.05. The figures were created by GraphPad Prism (version 7).

## 5. Conclusions

In conclusion, a diet supplemented with PQQ demonstrates beneficial effects in enhancing intestinal mucosal barrier function. The protective effects of PQQ on the intestine are likely attributed to the reduced level of intestinal proinflammatory cytokines and cell apoptosis via the inhibition of the MKK3/6-p38 pathway.

## Figures and Tables

**Figure 1 ijms-25-09723-f001:**
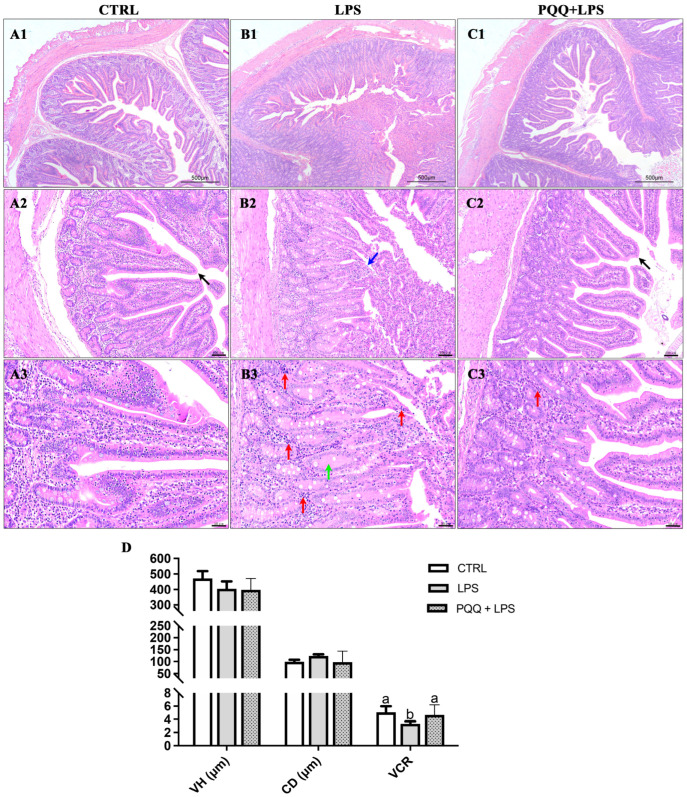
Effects of PQQ supplementation on jejunal morphology in weaned piglets challenged with LPS. (**A1**–**A3**) represent HE staining of jejunal tissue from the CTRL group; (**B1**–**B3**) represent jejunal tissue from the LPS group; (**C1**–**C3**) represent jejunal tissue from the PQQ + LPS group. The scale bars are as follows: (**A1**,**B1**,**C1**) 500 μm, (**A2**,**B2**,**C2**) 100 μm, and (**A3**,**B3**,**C3**) 50 μm. (**D**) The villus height, crypt depth, and villus height and crypt depth ratio among the CTRL, LPS, and PQQ + LPS groups. Black arrows indicate healthy and intact villi. Blue arrows represent epithelial detachment in the mucosal layer exposing the lamina propria. Red arrows show inflammatory cell infiltration characterized by densely packed cells with deeply stained nuclei. Green arrows indicate edematous fluid in the crypts, appearing as lightly stained areas. The labels ‘a’ and ‘b’ represent the outcomes of multiple comparisons conducted on the data. Groups sharing the same letter do not differ significantly from each other, while groups with different letters exhibit statistically significant differences (*p* < 0.05). CTRL piglets received a basal diet and were injected with 0.9% of NaCl; LPS piglets received a basal diet and were challenged with LPS; LPS + PQQ, piglets received a 3.0 mg/kg PQQ diet and were challenged with LPS, *n* = 6.

**Figure 2 ijms-25-09723-f002:**
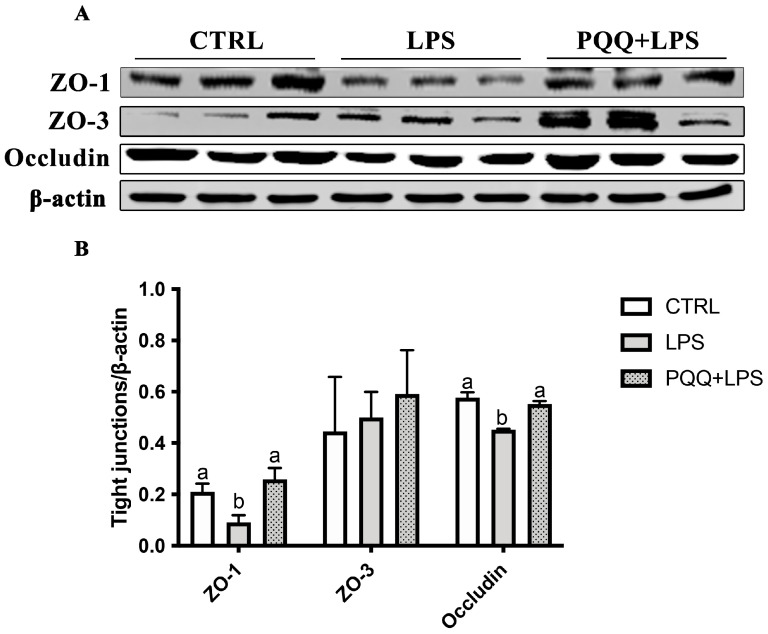
Effects of PQQ supplementation diet on jejunal mucosa tight junction proteins of weaned piglets challenged with LPS. (**A**) Protein levels of ZO-1, ZO-3, Occludin, and the densitometric values of tight junctions normalized to β-actin; (**B**) Statistical quantification for the data in A, *n* = 6. The labels ‘a’ and ‘b’ represent the outcomes of multiple comparisons conducted on the data. Groups sharing the same letter do not differ significantly from each other, while groups with different letters exhibit statistically significant differences (*p* < 0.05). CTRL piglets received a basal diet and were injected with 0.9% of NaCl; LPS piglets received a basal diet and were challenged with LPS; LPS + PQQ piglets received a 3.0 mg/kg PQQ diet and were challenged with LPS.

**Figure 3 ijms-25-09723-f003:**
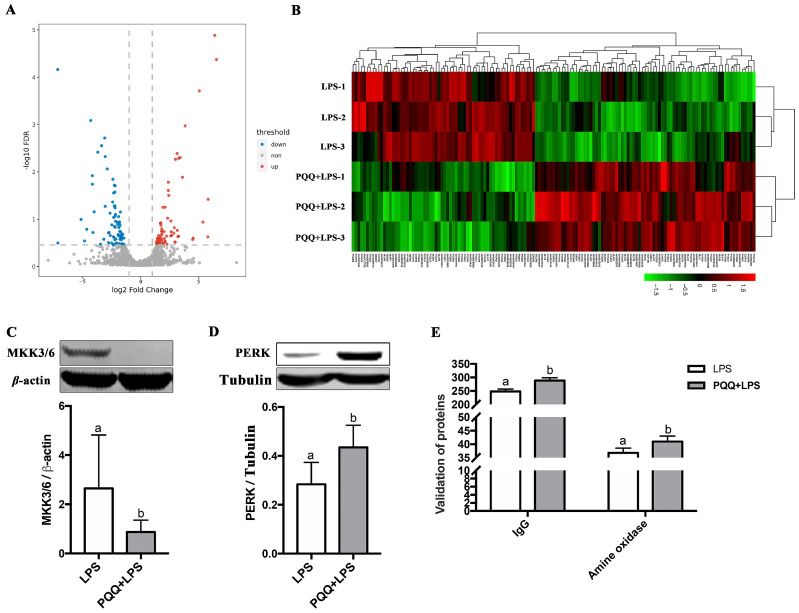
Quantitative analysis and validation of proteomics data for the LPS group and the PQQ + LPS group. (**A**) Volcano map: the red dots represent up-regulated proteins, while the blue dots represent down-regulated proteins. Proteins that showed no difference were represented by gray dots; (**B**) Heat map: red and green represent the high and low expression proteins; (**C**,**D**) Verification of MKK3/6 and PERK level in the LPS group and the PQQ + LPS group by Western blot; (**E**) The expression of IgG and amine oxidase levels in the LPS group and the PQQ + LPS group were verified by ELISA. The labels ‘a’ and ‘b’ represent the outcomes of multiple comparisons conducted on the data. Groups sharing the same letter do not differ significantly from each other, while groups with different letters exhibit statistically significant differences (*p* < 0.05). LPS piglets received a basal diet and were challenged with LPS; LPS + PQQ piglets received a 3.0 mg/kg PQQ diet and were challenged with LPS, *n* = 3.

**Figure 4 ijms-25-09723-f004:**
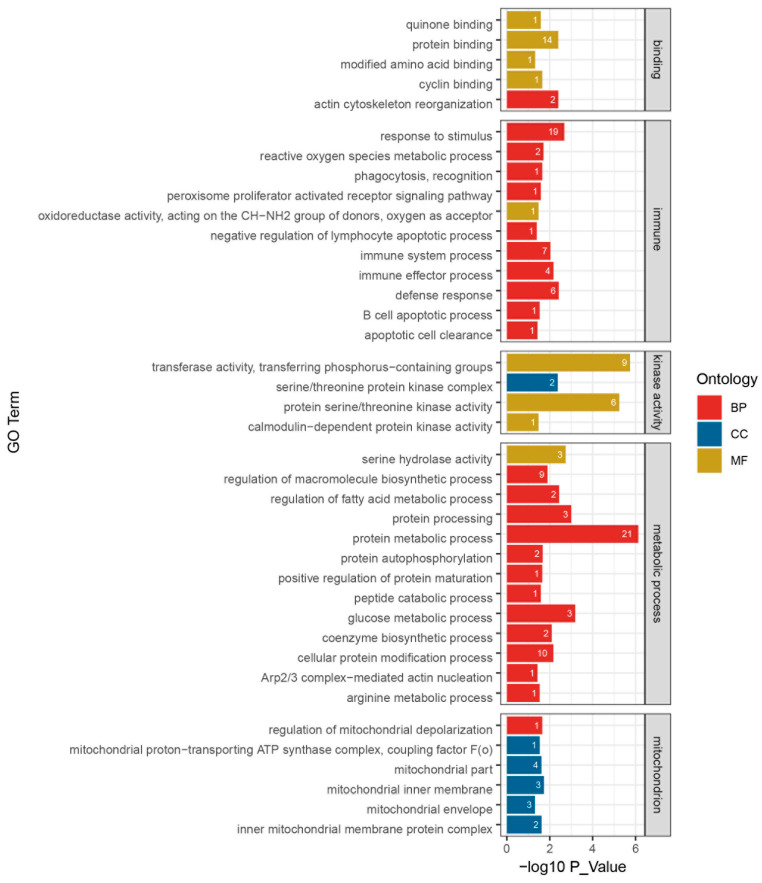
Histogram for GO enrichment and classification of differential proteins for the LPS group and the PQQ + LPS group. The ordinate represents the GO term, and the horizontal axis represents −log_10_P value. The number at the top of the column denotes differences in protein numbers in the GO term. Metabolic process: GO terms related to differential enrichment are associated with metabolism. Mitochondrion: GO terms related to differential enrichment are associated with mitochondria. Immune: GO terms related to differential enrichment are associated with immune functions. Binding: GO terms related to differential enrichment are associated with various molecular binding activities. Kinase activity: GO terms related to differential enrichment are associated with various kinase activities. BP, biological process. CC, cellular component. MF, molecular function, *n* = 3.

**Figure 5 ijms-25-09723-f005:**
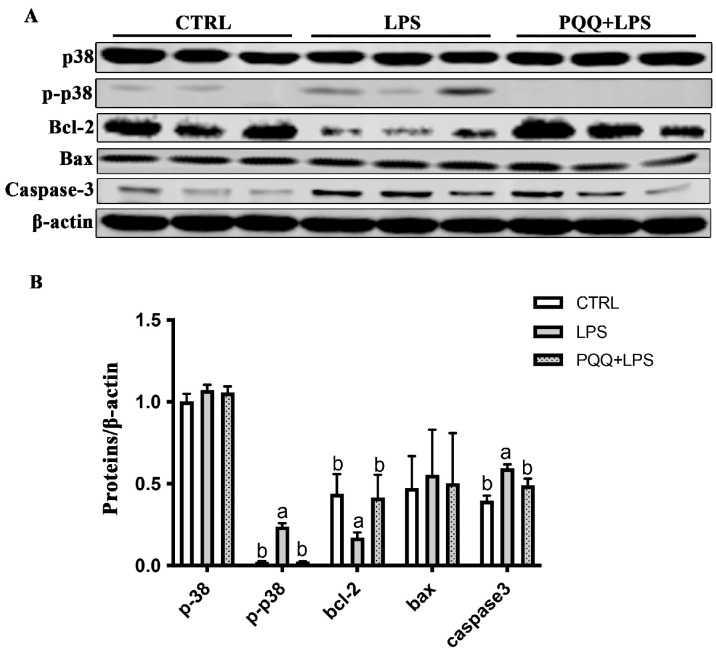
Effects of PQQ supplementation diet on expression of p38 and apoptosis-related protein levels in weaned piglets challenged with LPS. (**A**) The densitometric values of proteins normalized to β-actin expression; (**B**) Statistical quantification of the data in picture A. The labels ‘a’ and ‘b’ represent the outcomes of multiple comparisons conducted on the data. Groups sharing the same letter do not differ significantly from each other, while groups with different letters exhibit statistically significant differences (*p* < 0.05). *n* = 6. CTRL piglets received a basal diet and were injected with 0.9% of NaCl; LPS piglets received a basal diet and were challenged with LPS; LPS + PQQ piglets received a 3.0 mg/kg PQQ diet and were challenged with LPS.

**Figure 6 ijms-25-09723-f006:**
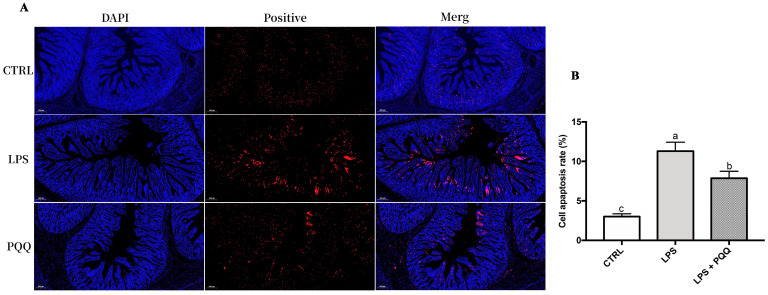
PQQ supplementation diet reduced LPS-induced apoptosis on jejunum (**A**) DAPI, jejunal tissue nucleus staining (blue); positive, fluorescence staining for TUNEL positive cells (red) of jejunum; the scale bar represents 200 μm. CTRL piglets received a basal diet and were injected with 0.9% of NaCl; LPS, piglets received a basal diet and were challenged with LPS; LPS + PQQ, piglets received a 3.0 mg/kg PQQ diet and were challenged with LPS; (**B**) Statistical quantification of the data in picture A. The labels ‘a’, ‘b’, and ‘c’ represent the outcomes of multiple comparisons conducted on the data. Groups sharing the same letter do not differ significantly from each other, while groups with different letters exhibit statistically significant differences, *p* < 0.05, *n* = 6.

**Table 1 ijms-25-09723-t001:** Effects of PQQ supplementation on growth performance and diarrhea index in weaned piglets challenged with LPS ^1^.

Item	CTRL	LPS	PQQ + LPS	SEM	*p* Value
Initial BW (kg)	7.16	7.16	7.16	0.85	0.96
Final BW (kg)	11.79 ^a^	10.05 ^b^	10.48 ^ab^	1.14	0.03
ADG (g)	330.00 ^a^	237.00 ^b^	206.00 ^b^	0.05	<0.01
ADFI (g)	575.00	463.00	445.00	0.10	0.08
FCR	1.76	2.35	1.92	0.62	0.33
Diarrhea index (%)	1.98 ^b^	7.54 ^a^	4.37 ^ab^	0.33	0.02

^1^ BW, body weight. ADG, average daily gain. ADFI, average daily feed intake. FCR, feed conversion ratio. CTRL piglets received a basal diet and were injected with 0.9% of NaCl; LPS piglets received a basal diet and were challenged with LPS; LPS + PQQ piglets received a 3.0 mg/kg PQQ diet and were challenged with LPS. The labels ‘a’, ‘b’, and ‘ab’ represent the outcomes of multiple comparisons conducted on the data. Groups sharing the same letter do not differ significantly from each other, while groups with different letters exhibit statistically significant differences (*p* < 0.05). If ‘ab’ appears, it indicates no significant difference compared to both groups labeled ‘a’ and ‘b’. *n* = 6.

**Table 2 ijms-25-09723-t002:** Effects of PQQ supplementation on cytokines in serum and mucosa of weaned piglets challenged with LPS ^1^.

Items	CTRL	LPS	PQQ + LPS	SEM	*p* Value
sDAO (U/mL)	4.95 ^c^	6.02 ^a^	5.73 ^b^	0.06	<0.01
sI-FABP (pg/mL)	200.06 ^b^	234.82 ^a^	210.76 ^b^	5.58	<0.01
sD-LA (nmol/mL)	13.27 ^b^	14.40 ^ab^	14.95 ^a^	0.37	0.02
sALP (U/L)	336.61 ^a^	224.97 ^b^	209.52 ^b^	18.06	<0.01
mI-FABP (pg/mg)	8.03	9.37	8.41	0.42	0.12
mD-LA (nmol/mg)	0.86 ^a^	1.01 ^b^	1.02 ^b^	0.04	0.03
mTNF-α (pg/mg)	18.69 ^b^	23.98 ^a^	20.60 ^b^	0.77	<0.01
mTGF-β1 (pg/mg)	15.59	15.20	13.79	1.46	0.67
mIFN-γ (pg/mg)	9.36 ^b^	12.23 ^a^	10.85 ^ab^	0.47	<0.01
mIL-1β (pg/mg)	5.77	6.39	6.53	0.31	0.23

^1^ The initial “s” in the abbreviation represents specific serum-related indicators or analyses. The initial “m” denoted in the abbreviation represents specific jejunal mucosa-related indicators or analyses. CTRL piglets received a basal diet and were injected with 0.9% of NaCl; LPS piglets received a basal diet and were challenged with LPS; LPS + PQQ piglets received a 3.0 mg/kg PQQ diet and were challenged with LPS. The labels ‘a’, ‘b’, ‘c’, and ‘ab’ represent the outcomes of multiple comparisons conducted on the data. Groups sharing the same letter do not differ significantly from each other, while groups with different letters exhibit statistically significant differences (*p* < 0.05). If ‘ab’ appears, it indicates no significant difference compared to both groups labeled ‘a’ and ‘b’. *n* = 6.

**Table 3 ijms-25-09723-t003:** The differentially expressed proteins were subjected to KEGG pathway enrichment analysis ^1^.

KEGG Term (*p* Value)/Protein Name	Gene Name	Protein ID in UniProt	Log_2_FC
Antigen processing and presentation (*p* = 0.002)			
MHC I6	SLA-6	A1YH88	1.53
MHC I2	SLA-2	A0A2S1PUH7	−4.65
Cathepsin L1	CTSL	Q28944	−1.61
Apoptosis (*p* = 0.01)			
IgG heavy chain	IGHG	L8B139	1.64
Poly-Ig receptor	unknow	Q9N2H7	5.79
Fc epsilon RI signaling pathway (*p* = 0.02)			
IgG heavy chain	IGHG	L8B139	1.64
MKK3/6	MAP2K6	F1RV28	−2
Inflammatory mediator regulation of TRP channels (*p* = 0.008)			
MKK3/6	MAP2K6	F1RV28	−2
Cytochrome P450 2C42	CYP2C42	I3LCZ8	−2.89
Calcium/calmodulin-dependent protein kinase	CAMK2G	A0A286ZKT1	−2.26
Intestinal immune network for IgA production (*p* = 0.01)			
IgG heavy chain	IGHG	L8B139	1.64
Poly-Ig receptor	unknow	Q9N2H7	5.79
Tight junction (*p* = 0.01)			
MAGUK p55 subfamily member 5	MPP5	F1SA41	3.15
Claudin	CLDN7	C3VPJ4	−1.59
PATJ crumbs cell polarity complex component	PATJ	A0A287AG82	−2.23

^1^ A *p* value of less than 0.05 indicates significant enrichment of the KEGG pathway entry between the LPS group and the PQQ + LPS group, *n* = 3. The value of Log_2_FC: A positive value indicates a significant increase in the differential-expressed protein within that signaling pathway in the LPS + PQQ group compared to the LPS group, whereas a negative value indicates a significant decrease in the LPS + PQQ group compared to the LPS group.

## Data Availability

Data is contained within the article and Appendix A.

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
