# Peer review of "Pyrroloquinoline Quinone Alleviates Intestinal Inflammation and Cell Apoptosis via the MKK3/6-P38 Pathway in a Piglet Model"

_ijms, 2024, doi:10.3390/ijms25179723_

Round 1
Reviewer 1 Report
Comments and Suggestions for Authors
Minor suitability points:
Introduction
- Requires better detailing of the PQQ. Characteristics ? For the journal in question, is information about the molecule studied as relevant as its biological functions?
Methodology
- How was the PQQ obtained and its purity level determined?
- Which post-tests are used in statistical analyses? Describe them.
Results
- Add in full the meaning of BW, in the results paragraph in 3.1.
-Results: for all tables it is necessary to add in the legend how the comparisons between the groups were made. What does “a”, “b”, “ab” mean? Describe to better understand your results.
- In histological images it is necessary to add images with greater amplification, as well as the addition of arrows identifying each of the characteristics evaluated. As presented, there is little information value in relation to the written results for histological evaluation. Subtitles are also required.
- Other figures still require greater detail to help interpret the results, which are often succinct.
Points of greatest attention:
Methodology
- Why only determine the jejunum in the assessments? Many studies address assessments with all intestinal segments. Justify the reason (theoretical and scientific basis) for choosing the jejunum and excluding the other segments.
- What parameters are used in histological evaluation? Did you use scores? How was the standardization of assessments? Was it double blind? Triple-blind? Without this information, such data can only be approached in a qualitative way and subject to interpretation bias by the pathologist.
Results
- From the title, the authors address the possible effects of PQQ through anti-inflammatory properties, however the results expressed, for example, in the table do not highlight this potential effect, for example, in the reduction of pro-inflammatory cytokines, which are key components in the genesis of the inflammatory process.
Despite the robust methodology used in the experiment, the writing and description of methodological procedures and results devalues the article as a whole. Extensive adjustments are necessary for a new assessment.
Reviewer 2 Report
Comments and Suggestions for Authors
In the manuscript by Huang et al., Pyrroloquinoline quinone alleviates intestinal inflammation and cell apoptosis via MKK3/6-P38 pathway in a piglet model, authors have evaluated mechanistic role of dietary supplement of PQQ in settings of LPS induced intestinal inflammation. It is a well-designed and executed study. Although study shows novelty, I have following concern:
1. What is rational to use LPS, it causes systemic inflammation. Better to use a different agent like DSS.
2. Authors have used 72 piglets. Please provide a table with number of animals for each experiment.
3. In many experiments, N=3 was used which is not sufficient to conclude any effect. I would recommend repeating the experiment to add more observations to get N=6 to draw proper conclusion. What was the primary outcome. How power calculation was done for primary outcome.
4. As LPS known to cause systemic inflammation also, what was the effect of PQQ on soluble/plasma cytokines
5. In figure 2, ZO-3 is increased by PQQ than CTRL, authors have any thoughts on that.
6. In some figures, N is missing in figure legends.
7. In figure 2, t-ERK should be estimated
Round 2
Reviewer 1 Report
Comments and Suggestions for Authors
After the necessary adjustments made by the authors, I consider it suitable for publication.
Author Response
Thanks for your professional suggestions and positive feedback.
Reviewer 2 Report
Comments and Suggestions for Authors
The authors have answered all comments. Although authors have explained for N=3, I think N=3 is not enough to conclude any effect.
Round 3
Reviewer 2 Report
Comments and Suggestions for Authors
Authors have explained the my previous concern (still I have concern highlighted in red text) as: we expanded the experiment to include additional replicates. We have conducted Western blot analyses on three more replicates and re-quantified the protein levels for ZO-1, ZO-3, occludin, p38, p-p38, Bax, Bcl-2, and caspase-3 (Did authors conducted new experiment for addition of more N=3 or used previous one?). The results for all protein expressions were consistent with our initial findings, except for ZO-3. After increasing the sample size, although ZO-3 levels in the LPS + PQQ treatment group remained higher compared to the control and LPS groups, ANOVA showed no statistically
significant difference. This change in significance could be attributed to several factors:
1. Increased sample variability: The inclusion of more samples likely introduced greater variability within groups, which may have masked the initial differences observed (Variability is one the key factor for human study should be consider while designing experiment).
2. Smaller effect size: The effect size for ZO-3 may be smaller than initially estimated. As the sample size increased, the true effect could have been diluted, resulting in a loss of statistical significance (I am not agree with this, that is the case than there might be error in power calculation for sample size).
Round 4
Reviewer 2 Report
Comments and Suggestions for Authors
No further comment.